# On the Nature of Functional Differentiation: The Role of Self-Organization with Constraints

**DOI:** 10.3390/e24020240

**Published:** 2022-02-04

**Authors:** Ichiro Tsuda, Hiroshi Watanabe, Hiromichi Tsukada, Yutaka Yamaguti

**Affiliations:** 1Chubu University Academy of Emerging Sciences, Chubu University, Aichi, Kasugai 487-8501, Japan; 2Center for Mathematical Science and Artificial Intelligence, Chubu University, Aichi, Kasugai 487-8501, Japan; hwata@isc.chubu.ac.jp (H.W.); tsukada@isc.chubu.ac.jp (H.T.); 3Faculty of Information Engineering, Fukuoka Institute of Technology, Fukuoka 811-0295, Japan; y-yamaguchi@fit.ac.jp

**Keywords:** self-organization with constraints, functional differentiation, hierarchy, heterarchy, nonstationarity, variational principle, superposition theorem

## Abstract

The focus of this article is the self-organization of neural systems under constraints. In 2016, we proposed a theory for self-organization with constraints to clarify the neural mechanism of functional differentiation. As a typical application of the theory, we developed evolutionary reservoir computers that exhibit functional differentiation of neurons. Regarding the self-organized structure of neural systems, Warren McCulloch described the neural networks of the brain as being “heterarchical”, rather than hierarchical, in structure. Unlike the fixed boundary conditions in conventional self-organization theory, where stationary phenomena are the target for study, the neural networks of the brain change their functional structure via synaptic learning and neural differentiation to exhibit specific functions, thereby adapting to nonstationary environmental changes. Thus, the neural network structure is altered dynamically among possible network structures. We refer to such changes as a dynamic heterarchy. Through the dynamic changes of the network structure under constraints, such as physical, chemical, and informational factors, which act on the whole system, neural systems realize functional differentiation or functional parcellation. Based on the computation results of our model for functional differentiation, we propose hypotheses on the neuronal mechanism of functional differentiation. Finally, using the Kolmogorov–Arnold–Sprecher superposition theorem, which can be realized by a layered deep neural network, we propose a possible scenario of functional (including cell) differentiation.

## 1. Introduction

The purpose of this study is to create a mathematical framework of functional differentiation using numerical analyses and a superposition theorem. In this paper, we propose hypotheses concerning the neural mechanism of functional (including cell) differentiation. According to the research based on the framework of self-organization with constraints [1], we developed a constrained self-organization theory and applied it to the interacting systems with complex environments, which were realized by new artificial intelligence, such as evolutionary reservoir computers (ERC) [2] and evolutionary dynamical systems (EDS) [3]. With certain constraints, ERC and EDS revealed the emergence of the network function via an optimized evolutionary process, which was associated with the self-organization of neuronal components (i.e., elements), thus leading to the realization of functional differentiation. The evolved structure of neural networks is discussed in relation to “heterarchical networks,” as proposed by Warren McCulloch, as the typical network structure of the brain [4]. In Section 2, we briefly review the conventional theory of self-organization and self-organization with constraints. In Section 3, we describe the heterarchical structures in comparison with hierarchical structures and uniform structures. In Section 4, we demonstrate that ERC and EDS yield heterarchical structures that change dynamically in the process of performing given tasks, thus suggesting that the brain is characterized as a “dynamic heterarchy.” In Section 5, based on the computation results of ERC and EDS, and the Kolmogorov–Arnold–Sprecher (KAS) superposition theorem, we propose a possible scenario for the functional differentiation of neurons or neural assemblies. Section 6 is a summary and outlook.

## 2. The Difference between Self-Organization and Self-Organization with Constraints

As discussed previously [1], scientific (not philosophical) studies of self-organization started in accordance with the cybernetics movement, in which the theory of self-organization developed in the construction of control theory [5]. Thereafter, in physics and chemistry, Haken and Prigogine et al. developed the concept of self-organization and formulated it as the emergence of macroscopic spatiotemporal patterns in far-from-equilibrium systems under stationary conditions. In fact, Prigogine et al. formulated self-organization phenomena observed particularly in chemical reactions and hydrodynamic systems, in terms of the variational principle of entropy production minimum [6]. As energy dissipation is a prerequisite in far-from-equilibrium systems, the concept of entropy flow associated with energy dissipation was introduced. In this respect, entropy production is defined as the sum of the change in the internal entropy of the system and the outflow of entropy from the system to the environment. Haken extended equilibrium phase transitions to far-from-equilibrium systems, introducing the slaving principle [7]. Moreover, Haken extended the Ginzburg–Landau formula to far-from-equilibrium and multicomponent systems. Although many modes appear in each critical point of transition, a few modes enslave other modes, which implies the appearance of order parameters from randomly interacting modes, including external noise.

Self-organizing phenomena are characterized by the appearance of macroscopic-ordered motion via cooperative and/or competitive interactions between the microscopic components of the system, namely atomic- or molecular-level interactions. The theory was successful in describing the phenomena observed in stationary and far-from-equilibrium states, for example, of target patterns, spiral patterns, propagating waves, and periodic and chaotic oscillations in many chemical and physical systems, such as chemical reaction systems, hydrodynamic systems, optical systems, and geophysics (see, e.g., [6,7]).

Another aspect of self-organization was highlighted in typical communication problems, as the brain activity in each communicating person may change according to individual intention factors and environmental factors, such as the purpose of the communication (see, e.g., [8,9,10,11]). This aspect can be formulated within a framework of functional differentiation [1], in which the functional elements (or components or subsystems) are produced by a certain constraint that acts on the whole system, based on the fact that neuronal functional differentiation occurs not only via genetic factors, but also via dynamic interactions between the brain and the environment [12,13,14,15]. Pattee [16] treated constraints by discriminating them from dynamics. He argued that a constraint is imposed via a rate-independent process, which is irreducible to dynamics; thus, it can control the system dynamics, which act as a rate-dependent process. We introduced a similar idea about constraints using a variational principle but treated the system dynamics that interacted with the environment as another constraint.

## 3. Dynamic Heterarchy

As discussed above, the brain is a self-organizing system with both internal and external constraints, thus yielding the dynamically nonstationary activity of neural networks. What type of self-organizing system is the brain? From the aspect of the variation of individual preferences, McCulloch considered an inconsistency in preference: people may prefer A to B, B to C, and C to A, which cannot be described by simple hierarchical values. However, this can be represented using a complex network. In fact, the brain solves such an inconsistency using the neural network. Furthermore, the brain solves another inconsistency, i.e., a circular causality: event A is caused by event B, B is caused by C, and C is caused by A. As a system that can handle these circular relations, McCulloch suggested the concept of heterarchy. Heterarchy is constructed by a top-down hierarchically connected system with additional bottom-up connections, in which a connection is defined as a sense of value and/or meaning, but not necessarily at a hardware level.

Considering McCulloch’s idea of heterarchy in the brain, Cumming emphasized the importance of this concept in wider fields, including social, economic, and ecological systems, and classified the systems into four types [17]: independent (nonrelational), uniform, hierarchical, and heterarchical systems (see Figure 1).

The neural networks in the brain change dynamically through learning mechanisms based on synaptic plasticity, depending on internal and external constraints. Because neurons in a higher-layered network of feedforward neural networks represent a higher function, it looks like a hierarchical system. Considering semantic factors, such as the values discussed by McCulloch, a hierarchical system demands the condition that there should be a semantic action from higher layers to lower layers of the network, which can be realized by sequential feedback connections from a higher to a lower layer. The hierarchical system of this sense can be observed in deep neural networks, following deep learning with the back-propagation algorithm. Another hypothesis proposes that hierarchically organized modular networks are evolved under the constraint of minimization of connection costs [18]. In contrast, actual brain networks typically consist of feedforward networks with feedback connections from a higher layer to multiple lower layers, whereby a simple hierarchy of function seems to be difficult to organize (see, e.g., [12,15]). In particular, the recurrent connections that occur in the brain could render the network architecture heterarchical, as stated by McCulloch. However, neuroscientists have distinguished between forward (ascending) and backward (descending) connections that have distinct anatomical and physiological properties, which may lead to distinct cognitive properties. This distinction has been used to integrate brain areas into a traditional concept of hierarchy [19,20], even in a dynamic phase [21]. However, a sufficient number of violations of a hierarchical designation exist: a small-world network, for example, which is actually observed in the cortex, makes it difficult to render the organization of brain areas as hierarchical in a traditional sense [22].

Therefore, casting the idea of a “heterarchical” organization in the brain will be valuable for future studies of “hierarchical” organization. Furthermore, the neural systems develop continually, caused by changes in the network structure via synaptic plasticity, dynamic interactions between neurons and glial cells, and interactions with the environment. These environmental interactions are nonstationary, because of the indeterminacy of the environment.

## 4. Heterarchy in the Model of Functional Differentiation

The developmental process of the brain occurs not in stationary states but in nonstationary states in far-from-equilibrium systems, typically yielding functional differentiation. Here, stationary states are defined as an unchanged probability distribution, and far-from-equilibrium systems are open systems, in which the states cannot reach the equilibrium states but could reach steady (unchanged in time), periodic, and chaotic states (see, e.g., [6,7]). Furthermore, nonstationary dynamics is also responsible for functional parcellation via functional connectivity, which changes according to the tasks and purposes [23]. Stationary states can be realized by fixed boundary conditions or initial conditions, whereas for the realization of nonstationary states, these conditions are not fixed, and a variational constraint that acts on the whole system is adopted. Therefore, in the latter case, the aim of the research is to identify an appropriate boundary and/or initial condition among various conditions to accomplish the given purposes. In fact, we studied developmental networks, including an ERC [2] and an EDS [3], and observed the network realization of functional differentiation and functional parcellation. Here, we demonstrate the appearance of the heterarchical structures of the networks in their developmental process.

We studied the evolution of a network of elementary dynamical systems and/or neuronal units, motivated by the following questions: 1. How did neuronal cells evolve and what are their roles in biological systems? 2. What is the relationship between the heterarchical network structures established in biological evolution and functional differentiation?

### 4.1. Heterarchy in EDS

We considered a network system of cells with functions that are not determined in advance, such as stem cells. The state of each cell, which is the network element, can be modeled by dynamical systems, i.e., $\left(\phi_{\lambda }^{t},\;\Omega \right)$, where Ω is a phase space and $\phi_{\lambda }^{t}$ is a parameter (λ)-dependent group action acting on each point in the phase space. This is called a dynamical rule, because it describes state transitions, or flow, because it integrates vector fields. In general, each dynamical system is designated by multiple parameters. Here, we assume that the states of different cells are assigned by different values of the parameter set. Given a constraint, for example, the maximum transmission of information, we introduced a genetic algorithm that changes a set of parameters to accomplish the constraint, thereby changing dynamical systems adaptively. As an elementary dynamical system in the network, we chose the following discrete-time dynamical system, where t is a nonnegative integer:(1)$$x_{k}\left(t+1\right)=a_{1}\tanh\left(a_{2}\left(x_{k}\left(t\right)-a_{3}\right)\right)-a_{4}\tanh\left(a_{5}\left(x_{k}\left(t\right)-a_{6}\right)\right)+b_{k}$$
For N-coupled dynamical systems, a set of parameters $\mathrm{GA}=(a_{1},\;\cdots ,\;a_{6},\;b_{1},\;\cdots ,\;b_{N}$) was considered a gene for the genetic algorithm. In the product space of the phase space and the parameter space, the overall dynamical systems can be represented by a coupled-dynamical system in various coupling manners, in which each dynamical rule was described by Equation (1), which produces monotone functions and unimodal and bimodal functions.

When introducing the maximum transmission of mutual information measured between a given external signal and each elementary dynamical system in the network, we found that each elementary dynamical system is differentiated to express an excitable spiking state, a passively susceptible state, and an oscillatory state, depending on the coupling strength of the network [3] (see Figure 2). This suggests that spiking neurons, oscillatory neurons, and glial active or passive responses were differentiated to satisfy the condition of interacting-cell systems that maximizes the transmission of external information. The overall system behaviors were weakly chaotic states, thereby allowing the acceleration of information transmission (see [24,25]). Complex structures via tripartite synapses between neurons and astrocytes were not explicitly found in the present computational model, because the model was organized to yield dynamical systems and their behaviors under the fixed network structure. Because tripartite synapses induce neuronal spiking and oscillations via calcium propagation [26], such a complex structure will be a target for the study within the framework of a certain optimization.

From this informational aspect of the network values in McCulloch’s sense, we investigated the change in the network structure (see Figure 3). For the first time, we constructed a randomly connected neural network that consisted of the elementary units described by Equation (1). We defined a layer of the network, according to the closeness to the unit 0, where closeness is defined as the least number of steps along the connected paths from unit 0 (Figure 3a). We applied the genetic algorithm to the connection strength under the fixed network topology (Figure 3b). In Figure 3b–d, the red (blue) directional lines indicate excitatory (inhibitory) connections, with the shades of color indicating the connection strength. In Figure 3c,d, we demonstrate the effect of information transmission after erasing the feedback connections: in Figure 3c, all feedback connections from higher to lower layers were erased, and in Figure 3d, only the feedback connection from unit 7 to unit 3 was erased. In the former case, the mutual information between unit 3 and the input decayed faster than that observed in the latter case. In contrast, in the latter case, information was preserved qualitatively, although the information quantity decreased slightly. It is noted that the mutual information between unit 3 and the input in the case without all feedback connections (Figure 3c) was larger than that in the case with a few feedback connections (Figure 3d). This is because information quantity can be supplied via feedforward connections rather than feedback connections.

In conclusion, the results of these numerical analyses imply that the input information can be preserved by the feedback connections and that the reduction of information quantity depends on the valance between feedforward and feedback connections. This is an important characteristic of heterarchical networks. In this model, the temporal change in mutual information conveys the values of input patterns. These numerical analyses suggest that a heterarchical structure is relevant for the effective preservation of the quality (values) of input information within the network system.

Based on the abovementioned numerical analyses, we propose a hypothesis for neuronal differentiation.

**Hypothesis** **1.**
*In biological evolution, neurons and glial cells evolved as functional elements, the coupled system of which is realized by tripartite synapses, and the networks of their functional elements can obtain external information and propagate it most effectively inside the system, as the coupled system can coordinate the overall chaotic system into synchrony of neuronal firing via the calcium waves in astrocytes [26]. The evolved network system includes a heterarchical structure. Moreover, the chaotic behaviors of the overall system accelerate the evolutionary development (see [24,25]).*


### 4.2. Heterarchy in ERC

A reservoir computer (RC) can be considered a model of the cerebral cortex [27,28,29] and the cerebellar cortex [30] because it is constructed by neural networks, including randomly coupled recurrent connections that play a role in the information processing of external time series. RCs learn the time series by changing only the synaptic connections from internal recurrent networks to output neurons. Therefore, RCs are not appropriate for use as a model of the cerebral cortex when they are adapted to changeable recurrent connections, such as those of the hippocampal CA3, although this idea of fixed recurrent networks is useful for rapid computation in the case without learning [31].

Motivated by the presence of synaptic learning in recurrent networks in the cerebral cortex, we extended RCs to allow for a changeable network structure, by introducing a genetic algorithm to produce ERC [2]. The dynamics of the network consisting of N neurons is described by the following equation for each elementary unit k:(2)$$x_{k}\left(t+1\right)=\left(1-\alpha_{k}\right)x_{k}\left(t\right)+\alpha_{k}\tan h(\sum_{l}w_{kl}x_{l}\left(t\right)+w_{k0}+\sum_{l}w_{kl}^{\left(in\right)}I_{l}\left(t\right))+\sigma_{k}\left(t\right),$$
where $x_{k}\left(t\right)$ is a state of the kth neuron (unit) at time t, $w_{kl}$ are the weights of recurrent connections in the internal reservoir network, $w_{kl}^{\left(in\right)}$ are weights of connections from an input signal $I_{k}\left(t\right)$, and $\sigma_{k}\left(t\right)$ is a noise term. The *k*th output unit is given by
(3)$$y_{k}^{*}\left(t\right)=\sum_{l}w_{kl}^{*}x_{l}\left(t\right),$$
where the asterisk indicates the type of input pattern, such as visual or auditory patterns.

In addition to synaptic learning at the output layer, which is performed by a conventional RC, we introduced a framework for changes of meta-parameters by adopting the rewiring of synaptic connections in internal recurrent networks, which can be accomplished by an evolutionary (genetic) algorithm, including mutation and crossover. We used several spatial and temporal patterns as different inputs in the learning phase. At the convergent state of the network, sensory neurons were differentiated for each type of sensory input pattern. This was a result of self-organization of the elementary units in the network system under constraints, in which the specific function of each elementary unit emerges concomitantly with the emergence of the overall network function. The network structure was suggestive of an effective network architecture for information processing, which actually occurred in biological evolution. In fact, the convergent network structure displayed a feedforward network, including feedback connections, which is quite similar to the structure observed in the hippocampus [32] and in local (not global) networks in the cerebral neocortex [33].

Figure 4 illustrates the change in the network structure of the ERC’s internal network, which consists of an input and output layer. In Figure 4a, the initial connections between both layers and the initial connections inside an input and output layer, which were given randomly in both wiring topology and strengths, are shown on the left, whereas the evolved connections are shown on the right. The feedforward connections were strengthened, and the feedback connections were weakened, which seems to indicate the appearance of a hierarchical structure; however, the number of feedback connections was increased, suggesting the appearance of a heterarchical structure. To clarify the emergence of such a structure, we computed the percentage of correct outputs of ERC with s times the weights of all feedback connections after the convergence of the network evolution of ERC, which is shown in Figure 4b. The abscissa indicates the scale factor of the weights, s, and the ordinate indicates the percentage of correct outputs that represent the accuracy of the functional differentiation of spatial (i.e., visual) and temporal (i.e., auditory) neurons, measured in relation to the case of originally evolved network weights. Here, the weights of the connections from the neurons in this internal output layer to the output neurons were changed again using the same Ridge regression algorithm. The computation results revealed that the accuracy of the functional differentiation tended to decrease when the scale factor of the weights changed from s=1. Furthermore, one can measure the energy consumption using a degree of connectivity, such as the number of synaptic connections or the overall strength of synaptic connections.

Considering these numerical analyses and other works [24,25,34,35,36,37,38], we propose the following hypotheses.

**Hypothesis** **2.***The functional differentiation for neuronal specificity, such as responding to specific external stimuli, evolved to minimize errors, which suggests the maximization of the transmitted information while reducing energy consumption*.

The developed ERC showed evolution to a heterarchical structure of the internal neural network from a random network in RC, which includes a feedforward network accompanied by a feedback network. This numerical analysis suggests the evolution of the hippocampus from reptiles to mammals [14]. Because the mammalian hippocampus includes a feedforward network from the CA3 to the CA1, accompanied by a feedback network in the CA3, it is plausible to think that the formation of episodic memory can be realized by this kind of evolution of the heterarchical network structure.

**Hypothesis** **3.**
*In the biological evolution of the hippocampus from reptiles to mammals, mammals became able to associate different memories successively, because of their heterarchical structure, whereas reptiles could perform a single association of memory, because of their random networks. Therefore, episodic memory became possible in mammals.*


**Hypothesis** **4.***The neural networks required to yield functional differentiation are evolutionarily self-organized to exhibit a heterarchical structure via the appearance of feedback connections within an architecture composed of forward connections*.

## 5. Superposition Theorem, Epigenetic Landscape, and Functional Differentiation

Functional differentiation in the brain begins during embryogenesis and is completed during the developmental process and thus depends on both genetic factors and environmental factors, whereas functional parcellation is realized via the functional connectivity between neurons or neural assemblies in a task-dependent manner. Here, we focus on a supposedly common process that can be described by mathematical functions. Figure 5 illustrates the process of cell differentiation from pluripotent stem cells, which was schematically drawn using the landscape representation. A similar process can occur in functional differentiation. We asked how this differentiation process is represented by mathematical functions; to address this question, we referred to the superposition theorem proven by Kolmogorov, Arnold, et al. ([39,40,41]). We also asked how the change in landscape could represent cellular or functional differentiation; to address this question, we studied the changes in the network of dynamical systems using the transformation of indices, which represent transcription factors, for example.

### 5.1. The Kolmogorov–Arnold–Sprecher Superposition Theorem

The representation of multivariate continuous functions in terms of single-variable continuous functions is a challenging problem that has attracted the interest of many researchers, particularly in relation to the solvability of algebraic equations proposed by Hilbert (for example, see [42]). Kolmogorov, Arnold, and, later, Sprecher and others proved that such a representation was possible. Here, we consider the representation of an n-variable continuous function, f: [0, 1]n→ℝ, in terms of the superposition of the following continuous functions: ψ:[0, 2]→[0, 2], which is a (ν, α)-Hölder continuous (inner) function, and ϕj:[0, 2γ−1γ−2] →ℝ, indicating m+1 continuous (outer) functions. We call a function g:[a, b]→[a, b] (ν, α)-Hölder continuous if and only if there exist ν>0 and 0<α≤1, such that |g(x)−g(y)|≤ν|x−y|α for all x, y∈[a, b]. The case of α=1 indicates a ν-Lipschitz continuous function. Let n≥2, m≥2n, and γ≥m+2 be integers. One can determine real numbers, c, λi(i=1, ⋯n),ν, and α, in the following way:

$c=\frac{1}{\gamma \left(\gamma -1\right)},\;\lambda_{1}=1,\;\lambda_{i}=\sum_{l=1}^{\infty }\gamma^{-\left(i-1\right)\beta_{n}\left(l\right)}\;\left(2\leq i\leq n\right)$ with $\beta_{n}\left(l\right)=\frac{1-n^{l}}{1-n}$, and $\nu =2^{-\alpha }\left(\gamma +3\right),\;\alpha =\log_{\gamma }2$. These relations are necessary for the actual construction of the inner and outer functions.

**Theorem** **1.**
*(The Kolmogorov–Arnold–Sprecher (KAS) superposition theorem.)*

*Using the conditions described above, the following representation was obtained:*



(4)
$$f\left(x_{1},\;x_{2},\;\cdots ,\;x_{n}\right)=\overset{m}{\underset{j=0}{\sum }}\phi_{j}(\overset{n}{\underset{i=1}{\sum }}\lambda_{i}\psi \left(x_{i}+jc\right))$$


Proof was given in a constructive way that could lead us to the realization of ψ(x) by continuous but self-similar functions, which are, in particular, the monotonically increasing functions. The inner function ψ(x) is independent of $f\left(x_{1},\;x_{2},\;\cdots ,\;x_{n}\right)$. (refer to [42,43] for proof).

Regarding the realization of this theorem by neural networks, Hecht-Nielsen first identified its possibility [44]. Funahashi proved the approximate realization of multivariate continuous functions by neural networks by referring to the KAS theorem [45]. Recently, Montanelli and Young proved in a constructive way that deep rectified linear unit (ReLU) networks approximate any multivariate continuous functions by using the KAS theorem [42], where ReLU networks refer to the networks consisting of the units with a rectified linear activation function, as defined by the following equation:(5)g(z)=max(0,z),
where z is the summation of the inputs to each neuronal unit and g is an activation function.

### 5.2. Epigenetic Landscape with Indices of Phenotype Yielding Functional Differentiation

Is there any evidence that dynamical systems are related to the cell differentiation from neural stem cells? Imayoshi et al. [46] found that, in the mouse, the dynamic change in the activity of three kinds of transcription factors of the bHLH type, namely Ascl1, Hes1, and Olig2, contributes to the differentiation of neural stem cells to GABAergic neurons, astrocytes, and oligodendrocytes, respectively. These three proteins display oscillations in their concentrations with different periods. When these oscillatory states are maintained, self-reproduction of neural stem cells is facilitated. In contrast, when the concentration (which implies the expression level) of any of the proteins increases to inhibit the others, the corresponding cell differentiation occurs. For example, if the concentration of Ascl1 increases and inhibits the other two proteins, then differentiation to astrocytes and oligodendrocytes is inhibited and only neurons differentiate. Furthermore, Furusawa and Kaneko [47] pursued the dynamical mechanism of cell differentiation and succeeded in explaining the dynamics associated with cell differentiation with coupled dynamical systems and a fluctuation–dissipation theorem of the nonequilibrium type.

The realization of the theorem by deep neural networks and the facts described above led us to formulate the following working hypothesis for the dynamical mechanism of functional differentiation.

**Working** **hypothesis:***We assume that dynamical systems provide an underlying mechanism of functional differentiation**. Then, it is probable that differentiated neuronal characters are represented by attractors, yielding multistable states in the phase space of the neural network. Here, the phase space is a state space representing neural activity. A nondifferentiated state, such as the state of (neuronal) stem cells, has no attractor or only a single attractor, the basin of attraction of which is globally flat, thus potentially representing “resilience” (see Figure 5). Therefore, functional differentiation is considered the dynamical change of basin structure in the phase space*.

To represent this aspect, we used the landscape, which describes the derivatives or local differences of vector fields in the phase space.

**Figure 5 entropy-24-00240-f005:**
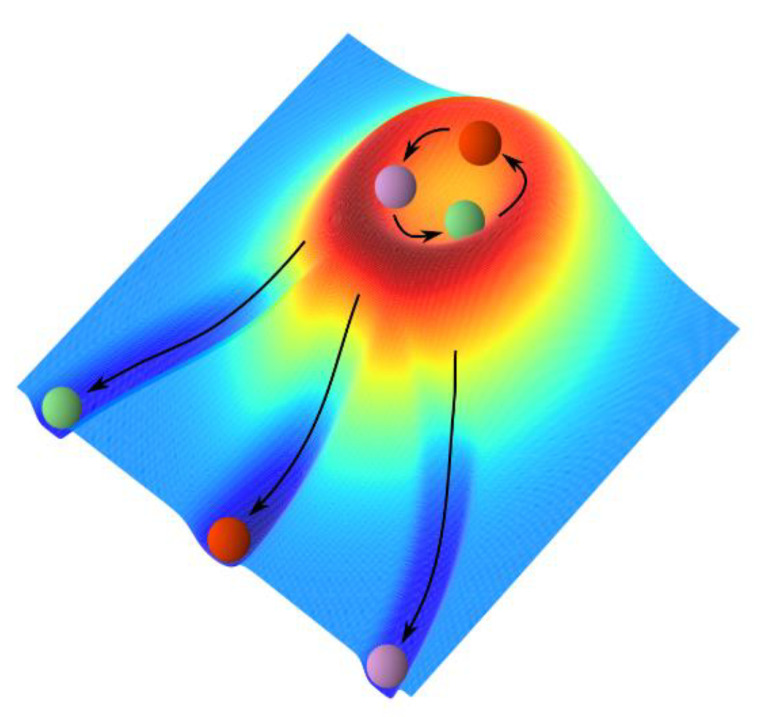
Schematic drawing of the mechanism of functional differentiation: three-dimensional landscape representation (see also [47]). The three colored balls represent three types of transcription factors that may behave in, for example, oscillatory states of concentration. The change in the landscape toward the left-lower part of the figure represents the differentiation of three types of neuronal cells that are realized via the mutual inhibition of transcription factors. Differentiation may be triggered by the inhibition triggered by one of the other factors. Here, neither the reprogramming (i.e., rejuvenation) nor the progenitor cell states shown in Waddington’s epigenetic landscape [48] are drawn.

The landscape with multiple stable states, i.e., multiple attractors, can be described by a higher-order continuous function, for example, an nth order polynomial function. However, the higher-order continuous functions are not necessarily differentiable, but simply a continuous function of variable y with n coefficients, $c_{1},c_{2},\cdots ,c_{n}$. This function, $D\left(y;\left\{c_{1},c_{2},\cdots ,c_{n}\right\}\right)$, can be viewed as an n-variable continuous function, f, as is represented in Equation (6):(6)$$f\left(c_{1},c_{2},\cdots ,c_{n}\right)\equiv D\left(y;\left\{c_{1},c_{2},\cdots ,c_{n}\right\}\right).$$
In contrast, the landscape with a single attractor can be described by, for example, the following equation:(7)$$S\left(y;\left\{b\right\}\right)=y^{n}+by+1.$$
If this landscape can be viewed as a function of a single parameter b, one obtains the following function, h:(8)h(b)≡S(y;{b}).
We interpreted function h(b) to be an activation state of a stem cell, such as a state of a transcription factor. S(y;{b}) was then interpreted to be a landscape of the phenotypic state of the stem cell using a transcription factor as an index of phenotype. Similarly, we interpreted function $f\left(c_{1},c_{2},\cdots ,c_{n}\right)$ to be based on a set of activation states of differentiated cells, such as a combination of transcription factors. Then, $D\left(y;\left\{c_{1},c_{2},\cdots ,c_{n}\right\}\right)$ was interpreted by a landscape of phenotypic states of differentiated cells using transcription factors as indices of phenotype. Based on the assumption described in the working hypothesis, each metastable state is represented by an attractor.

The application of the KAS theorem to the transformation from the state of a transcription factor in stem cells to a combination of transcription factors in differentiated cells allows the single-variable continuous function, $\psi \left(c_{i}\right),\;\left(i=1,2,\cdots ,n\right)$, which is an inner function of the theorem, to represent any n-variable continuous function $f\left(c_{1},c_{2},\cdots ,c_{n}\right)$ via composite functions with outer functions. Here, the theorem provides the transformation of the cells’ activation factors, such as transcription factors, which may play a role in the indices of phenotype. The corresponding cells’ states can be represented by the landscapes, $D\left(y;\left\{c_{1},c_{2},\cdots ,c_{n}\right\}\right)$, which change from S(y;{b}). Because $\psi \left(c_{i}\right),\;\left(i=1,2,\cdots ,n\right)$ can be viewed as the states of the transcription factors of stem cells, these should be independent of those of differentiated cells. This is compatible with the most important condition of the theorem. Figure 6 illustrates a single-variable function $\psi \left(c_{i}\right),\;\left(i=1,2,\cdots ,n\right)$ for various values of γ, the number of variables n, and the grid size k, whereby the minimum width, i.e., precision $\left(\gamma^{-k}\right)$ of drawing the graph of such a function, was given. When k→∞, the function is self-similar.

According to these interpretations, we propose the following hypothesis.

**Hypothesis** **5.***Functional differentiation is a process of selecting the initial conditions of neuronal stem cells, retrospectively, from which the development occurs toward an attractor that represents a differentiated cell as a target. Those initial conditions, which are certain parts of the basin of attraction in the landscape *S(y;{b})*, are designated by the continuous functions*$\psi \left(c_{i}\right),\;\left(i=1,2,\cdots ,n\right)$*. Another landscape of differentiated cells,*$D\left(y;\left\{c_{1},c_{2},\cdots ,c_{n}\right\}\right)$*, represents the activity state of phenotype, i.e., a combination of active proteins, which are designated by the multivariate continuous functions*$f\left(c_{1},c_{2},\cdots ,c_{n}\right)$.

The importance of environmental factors has been recognized in cell differentiation, such as factors that bring about reprogramming or rejuvenation. Reprogramming provides feedback information in the differentiation process. Similarly, in the functional differentiation of neural systems, the feedback information from environmental factors in the developmental process plays an important role in the adjustment of the self-organization of elementary units to that of an overall system. Therefore, the present theory provides a common framework for both cell and functional differentiation.

## 6. Summary and Outlook

In this paper, we proposed a mathematical framework of functional differentiation, based on the numerical results of both ERC and EDS and based on the KAS superposition theorem. We further proposed four hypotheses in relation to the following fundamental questions: (1) How do neuronal cells evolve and what are their roles? (2) What is the relationship between the heterarchical network structures established in biological evolution and functional differentiation? Furthermore, we asked how a change of landscape could represent cellular or functional differentiation. Using the working hypothesis, concerning the role of dynamical systems in the process of differentiation, we proposed another hypothesis about the epigenetic specification of cellular indices that produces functional differentiation. At this stage, these proposals remain advocative. However, further analyses according to the present theory of data, such as mRNA sequencing, are expected to clarify the relationship between the landscape and transcription factors and to justify an interpretation of functional differentiation in terms of the dynamic landscape.

## Figures and Tables

**Figure 1 entropy-24-00240-f001:**
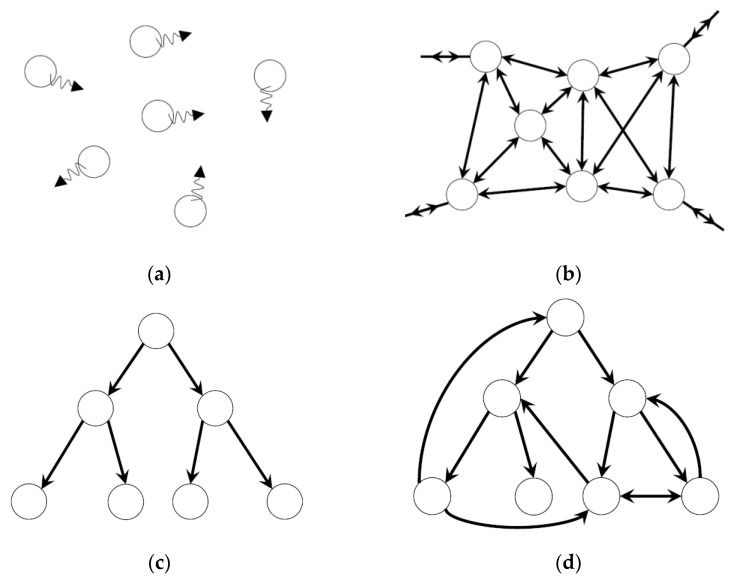
Four types of systems (see also Figure 2 in [17]): (**a**) independent; (**b**) uniform; (**c**) hierarchical; and (**d**) heterarchical.

**Figure 2 entropy-24-00240-f002:**
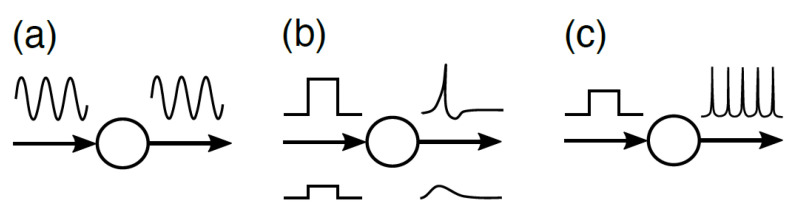
Schematic drawing of three typical differentiated states: (**a**) a passive state expressing a glial passive susceptible state, (**b**) an excitable state expressing a spiking neuron, and (**c**) an oscillatory state expressing an oscillatory neuron or glial oscillation (see [3] for numerical results of activity).

**Figure 3 entropy-24-00240-f003:**
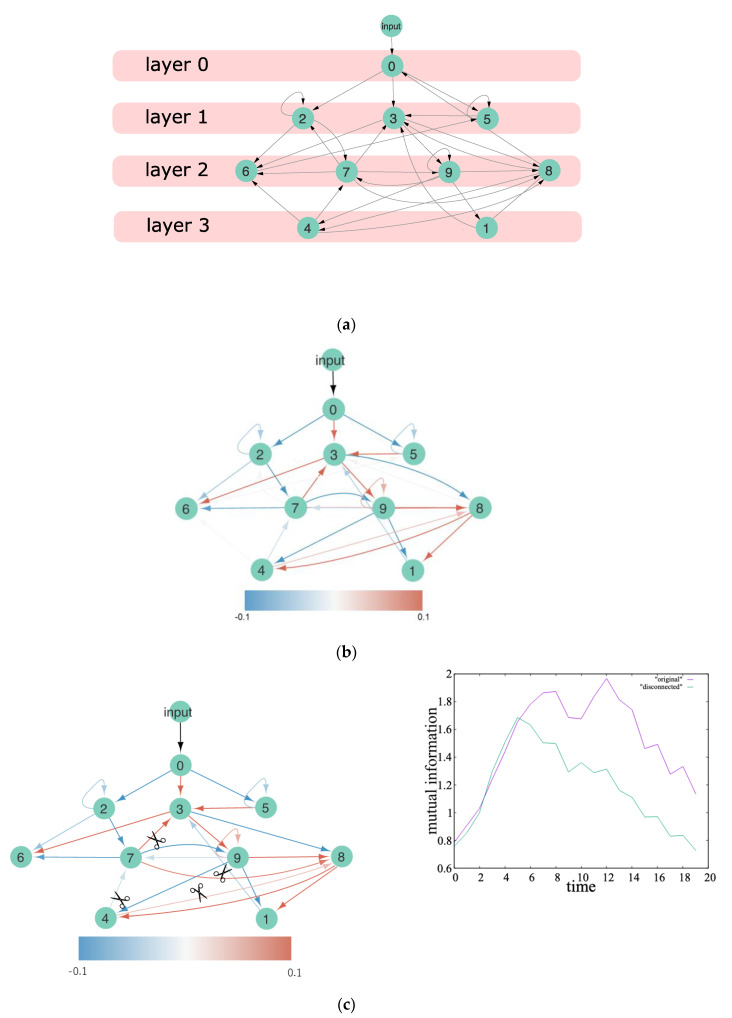
Numerical evidence of the emergence of a heterarchical structure in the evolved network of dynamical systems. (**a**) Randomly chosen network topology. Layers are defined by the closeness of units to the receiver unit, 0. In (**b**–**d**), the red (blue) directional lines indicate excitatory (inhibitory) connections, with the shades of color indicating the connection strength: the scale shown at a lower place of each figure. (**b**) Evolved network. Evolution was applied to the connection strength, preserving network topology. (**c**) Drastic reduction of information quantity in unit 3 after erasing all feedback connections (green curve in the panel on the right: the abscissa denotes time and the ordinate denotes mutual information between unit 3 and the input), in relation to the time change of information quantity in unit 3 of the original evolved network (a purple curve). (**d**) Change of information quantity in unit 3 after erasing only the feedback connection from unit 7 to unit 3 (green curve in the panel on the right). Information quantity was dropped, but qualitative behaviors did not change, which implies the preservation of quality (values) of information processing. Here, mutual information was calculated as time-dependent mutual information between two arbitrary units, which measures the dynamic change of shared information (see [24] for a detailed technique).

**Figure 4 entropy-24-00240-f004:**
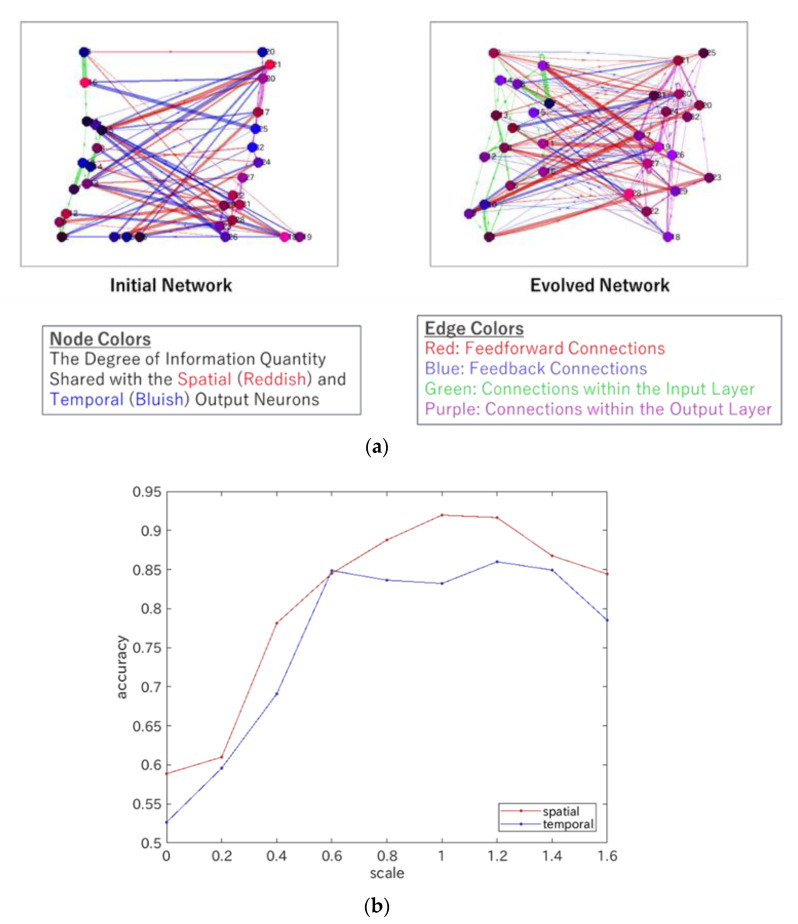
Numerical evidence of the heterarchical structure of an ERC. (**a**) Change in the internal network structure, consisting of an input and output layer from an initial random network to an evolved network. The network change proceeded with the change of wiring topology and connection weights, according to the optimization algorithm, such as the minimization of errors (present case, see [2] for ERC and [34,35] for predictive coding formulations), minimization of energy cost, or maximization of information (see, e.g., [3,18,24,25,36,37]). The colors of the nodes indicate the degree of information quantity shared with the spatial or temporal output neurons: the higher the shared information with spatial (temporal) output neurons, the deeper the reddish (bluish) color of the node. The colors of the edges are as follows: red for feedforward connections; blue for feedback connections; green for connections within the input layer; and purple for connections within the output layer. The thickness of the lines indicates the magnitude of the connection weights. (**b**) Change in the accuracy of the realization of functional differentiation with the change in the scale factor of feedback connections. The red (blue) curve denotes the accuracy of the spatial (temporal) neuron.

**Figure 6 entropy-24-00240-f006:**
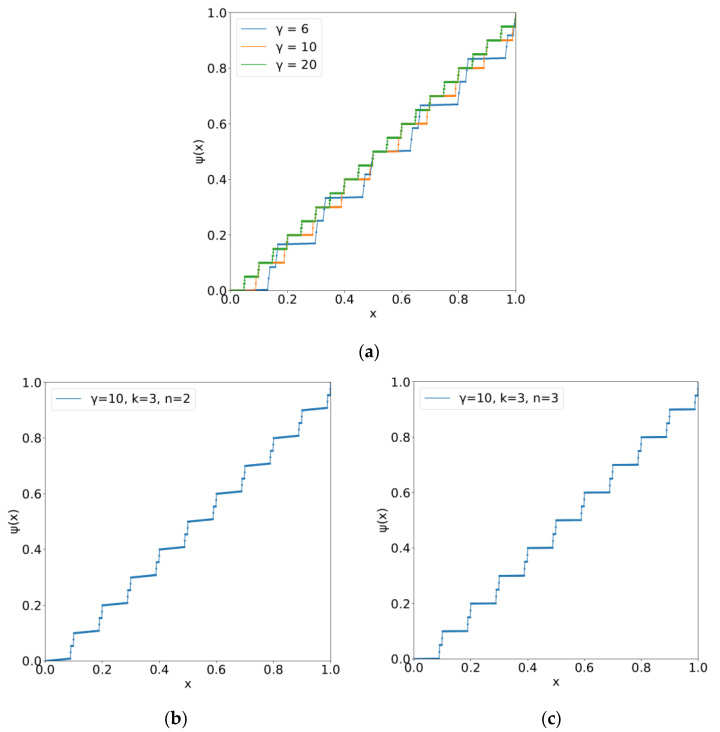
Numerical construction of the inner functions of Equation (4), *ψ*(*c_i_*), (*i* = 1,2,⋯,*n*). *k* = 3. An approximation of a single-variable function with a finite precision is shown, which can be an elementary function constituting a given *n*-variable function. In the present theory, this type of function is viewed as the states of transcription factors of stem cells. (**a**) *n* = 2. Blue, orange, and green indicate *γ* = 6, *γ* = 10, and *γ* = 20, respectively; (**b**) *n* = 2, *γ* = 10; (**c**) *n* = 3, *γ* = 10.

## Data Availability

Not applicable.

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
