# Peer review of "On the Nature of Functional Differentiation: The Role of Self-Organization with Constraints"

_entropy, 2022, doi:10.3390/e24020240_

Round 1
Reviewer 1 Report
Please see the attachment.

Author Response
Thank you very much for nice comments given by all of three reviewers, all of which were quite helpful for the revision of our manuscript.
The following is my reply to each comment.
Please see also an attached file that indicates the revised text explicitly.
For 1st Reviewer
(Reply) Concerning Pattee’s idea on constraints, I revised in the following way:
Pattee [16] treated constraints, discriminating from dynamics. He argued that a constraint is imposed via rate-independent process, which is irreducible to dynamics from outside the system, thus it can control the system dynamics acting as rate-dependent process, whereas dynamics is internally determined. We introduced a similar idea about constraints using a variational principle, but treated the system dynamics interacting that interacted with the environment as another constraint.
(Reply) According also to the 2nd Reviewer’s comment, I dropped “including people”.
(Reply) Yes, I have done.
(Reply) I recognize the importance of tripartite synapses between neurons and astrocyte, which may finally induce spiking and oscillations via the propagation of calcium. Fig.2 shows a clear computational example for neuronal differentiation, where complex structures such as tripartite synapses were not found. However, as this topic is important, I just added a brief comment on that in p.5:
The overall system behaviors were weakly chaotic states, thereby allowing the acceleration of information transmission (see [37]&[38]). Complex structures via tripartite synapses between neurons and astrocytes were not explicitly found in the present computational model, because the model was organized to yield dynamical systems and their behaviors under the fixed network structure. Because tripartite synapses induce neuronal spiking and oscillations via calcium propagation [36], such a complex structure will be a target for a future study within the framework of a certain optimization.
(Reply) Although the input in this model is fixed, the unit 3 is organized to receive feedback information from other higher-level of units, so that this feature matches the definition of heterarchy. In this article, we don’t treat coalition in the reviewer’s sense, although it is very interesting idea of coalition. I think that it is worth studying in future whether the present network system, in particular, ERC is realized as representing coalition.
(Reply) (vi)&(vii) According to the Reviewer’s comment, I changed the related part of text (p.6) in the following way:
In Fig. 3(c) and (d), we demonstrate the effect of information transmission after erasing the feedback connections: in (c), all feedback connections from higher to lower layers were erased, and in (d), only the feedback connection from unit 7 to unit 3 was erased. In the former case, the mutual information between unit 3 and the input decayed in time almost monotonously after faster than that observed in the latter case erasing the feedback connections. By contrast, in the latter case, information was preserved qualitatively, although the information quantity decreased slightly. It is noted that the mutual information between unit 3 and the input in the case without all feedback connections (Fig. 3 (c)) was larger than that in the case with a few feedback connections (Fig. 3(d)). This is because the information quantity can be supplied via feedforward connections rather than feedback connections.
In conclusion, the results of these numerical analyses imply that the input information can be preserved by the feedback connections, and that the reduction of information quantity depends on the valance between feedforward and feedback connections. This is an important characteristic of heterarchical networks.
(Reply) I added some sentences in Hypothesis 1 in the following way:
Hypothesis 1. In biological evolution, neurons and glial cells evolved as functional elements, the coupled system of which is realized by tripartite synapses, and the networks of their functional elements can obtain external information and propagate it most effectively inside the system, as the coupled system can coordinate the overall chaotic system into synchrony of neuronal firing via the calcium waves in astrocytes [36]. The evolved network system includes a heterarchical structure. Moreover, the chaotic behaviors of the overall system accelerate the evolutionary development (see [37–38]).

Reviewer 2 Report
This paper proposes a mathematical framework for the role of feedback mechanisms (as realised by heterarchical networks) in the process of cellular differentiation. This is based on a dynamical systems approach, where cellular states are represented by attractors in the potential landscape. This approach is likely to be relevant for future investigations. However, the current manuscript suffers somewhat from an imprecision of language, and sometimes lacks sufficient references to support the claims made. In the following, I will comment on some issues in more detail, as they appear while reading the manuscript.
Comments:
l.41: It is unnecessary to say ‘published in Entropy in 2016’.
ll.84-85 “… such as chemical reaction 84 systems, hydrodynamic systems, optical systems, and geophysics.” This should be properly referenced, or dropped.
ll.130-132 “In contrast, actual brain networks typically consist of feedforward networks with feedback connections from a higher layer of the network to multiple lower layers.” Again, a reference is necessary to follow where is this information from.
l.137 drop ‘including other people’. I think it is obvious that ‘other people’ are part of the environment.
l.145 “boundary conditions including initial conditions” which should be “boundary conditions or initial conditions”. Initial conditions are not boundary conditions.
ll. 153-155 The questions posed here are very grand questions, and too unspecific to be answered in a single publication. The questions can be made more specific by referring to the mathematical framework, for example.
ll.184-185 How do weakly chaotic states accelerate information transmission? A reference or an explanation is necessary here. (Also l.247)
Figure 3: How is information measured?
ll.272-274 Is this illustrated somewhere? Can you be more specific as to what kind of input patterns were used?
Hypothesis 2: How do you measure energy consumption in your neuronal networks? Is this quantified, and if yes, how?
Hypothesis 3: I do not see how this hypothesis can be inferred from the results presented previously. References are necessary to support the claims made about the evolution of the hippocampus.
Hypothesis 4: What is “valued” feedback?
ll.411-412 Doesn’t Figure 5 show a potential landscape? In this case, the vector field can be understood as the derivative of the potential landscape, not vice versa.
Figure 6: It is difficult to make out what is what here. The figure legends should be larger, and the axes should have labels.
Author Response
Thank you very much for nice comments given by all of three reviewers, all of which were quite helpful for the revision of our manuscript.
The following is my reply to each comment.
Please see also an attached file that indicates the revised text explicitly.
For 2nd Reviewer
This paper proposes a mathematical framework for the role of feedback mechanisms (as realised by heterarchical networks) in the process of cellular differentiation. This is based on a dynamical systems approach, where cellular states are represented by attractors in the potential landscape. This approach is likely to be relevant for future investigations. However, the current manuscript suffers somewhat from an imprecision of language, and sometimes lacks sufficient references to support the claims made. In the following, I will comment on some issues in more detail, as they appear while reading the manuscript.
Comments:
l.41: It is unnecessary to say ‘published in Entropy in 2016’.
(Reply) I deleted.
ll.84-85 “… such as chemical reaction 84 systems, hydrodynamic systems, optical systems, and geophysics.” This should be properly referenced, or dropped.
(Reply) I added typical two references.
ll.130-132 “In contrast, actual brain networks typically consist of feedforward networks with feedback connections from a higher layer of the network to multiple lower layers.” Again, a reference is necessary to follow where is this information from.
(Reply) I added typical two references.
l.137 drop ‘including other people’. I think it is obvious that ‘other people’ are part of the environment.
(Reply) Yes, dropped.
l.145 “boundary conditions including initial conditions” which should be “boundary conditions or initial conditions”. Initial conditions are not boundary conditions.
(Reply) I corrected in that way.
- 153-155 The questions posed here are very grand questions, and too unspecific to be answered in a single publication. The questions can be made more specific by referring to the mathematical framework, for example.
(Reply) I changed the sentences in the following way:
We studied the evolution of a network of elementary dynamical systems and/or neuronal units, motivated by We asked the following grand questions: 1. How did neuronal cells evolve and what are their roles in biological systems? 2. What is the relationship between the heterarchical network structures established in biological evolution and functional differentiation? To answer these questions, we studied the evolution of a network of elementary dynamical systems and/or neuronal units.
ll.184-185 How do weakly chaotic states accelerate information transmission? A reference or an explanation is necessary here. (Also l.247)
(Reply) I inserted two references.
Figure 3: How is information measured?
(Reply) I added a sentence in the legend of Fig.3:
Here, mutual information was calculated as time-dependent mutual information between two arbitrary units, which measures the dynamic change of shared information (see [37] for a detailed technique)
ll.272-274 Is this illustrated somewhere? Can you be more specific as to what kind of input patterns were used?
(Reply) To satisfy the comment, I added one sentence in l. 316-317:
We used several spatial and temporal patterns as different input in the learning phase.
Hypothesis 2: How do you measure energy consumption in your neuronal networks? Is this quantified, and if yes, how?
(Reply) I added one sentence in l. 344-346:
Furthermore, one can measure the energy consumption using a degree of connectivity such as a number of synaptic connections or an overall strength of synaptic connections.
Hypothesis 3: I do not see how this hypothesis can be inferred from the results presented previously. References are necessary to support the claims made about the evolution of the hippocampus.
(Reply) I added the following sentences before hypothesis 3:
The developed ERC showed evolution to a heterarchical structure of the internal neural network from a random network in RC, which includes a feedforward network accompanied by a feedback network. This numerical analysis suggests the evolution of the hippocampus from reptiles to mammals [14]. Because the mammalian hippocampus includes a feedforward network from the CA3 to the CA1, accompanied by a feedback network in the CA3, it is plausible to think that the formation of episodic memory can be realized by this kind of evolution of the heterarchical network structure.
Hypothesis 4: What is “valued” feedback?
(Reply) Deleted:
Hypothesis 4. The neural networks required to yield functional differentiation are evolutionarily self-organized to exhibit a heterarchical structure via the appearance of functional (or valued) feedback connections within an architecture composed of feedforward connections.
ll.411-412 Doesn’t Figure 5 show a potential landscape? In this case, the vector field can be understood as the derivative of the potential landscape, not vice versa.
(Reply) Figure 5 shows the landscape (not potential landscape) indicating active states of phenotype, which may be related to epigenetic landscape of Waddington.
Figure 6: It is difficult to make out what is what here. The figure legends should be larger, and the axes should have labels.
(Reply) In the legend of Fig. 6, I added some more explanation.
Numerical construction of the inner functions of Eq. (4), . = 3. An approximation of a single-variable function with a finite precision is shown, which can be an elementary function constituting a given-variable function. In the present theory, this type of function is viewed as the states of transcription factors of stem cells. (a) . Blue, orange, and green indicate , , and , respectively; (b) , ; (c) .
Thank you very much for nice comments given by all of three reviewers, all of which were quite helpful for the revision of our manuscript.
The following is my reply to each comment.
Please see also an attached file that indicates the revised text explicitly.

Reviewer 3 Report
I enjoyed reading this interesting use of the Kolmogorov-Arnold representation theorem to account for the epigenetic specification of neural networks and their functional differentiation. I do not fully understand your hypothesis; however, it sounds intriguing. I suspect that you will need to unpack this hypothesis using numerical simulations before other people will understand the idea. However, as a thought provoking hypothesis piece, I thought the current submission was viable.
I have a few comments that might help unpack some of your assertions. Perhaps you could consider the following:
Line 133: I would qualify the assertion that most network structures in the brain are hetero. Perhaps something like:
“One could argue that the recurrent nature of connectivity in the human brain renders the architecture heterarchical. However, people distinguish between forward (ascending) and backward (ascending) connections that have distinct anatomical and physiological properties. This distinction has been used to organise brain areas into a traditional hierarchy (Felleman and Van Essen 1991, Hilgetag, O'Neill et al. 2000). However, there are a sufficient number of violations of a hierarchical designation to render brain architectures heterarchical. Perhaps the best example of this are the frontal eye fields that occupy a high and low level in an archetypal hierarchy, depending upon the connections in question (Markov, Ercsey-Ravasz et al. 2013)."
Line 141: I think he you should define more clearly what you mean by stationary and far from equilibrium. I imagine that you meant something like:
"Non-stationarity in far from equilibrium systems refers here to itinerant dynamics over protracted timescales. Stationarity should be distinguished from steady state, in the sense that it is possible to have nonstationary itinerant dynamics in far from equilibrium systems with a steady-state solution. We mention this because what follows could be applied to (itinerant) systems that have a nonequilibrium steady-state solution."
Line 200: could you replace "we adopted" with "we applied".
Line 209: I would say: "These numerical analyses suggest that a heterarchical structure is relevant for the effective…"
Line 323: similarly, replace "computation results" with "numerical analyses".
Line 324: it might be useful to reference related hypotheses. For example, the minimisation of errors underwrites predictive coding formulations of message passing in the brain (Srinivasan, Laughlin et al. 1982, Rao and Ballard 1999). You might also refer to formulations in terms of information transfer and maximum mutual information (Barlow 1974, Optican and Richmond 1987, Linsker 1990).
Line 335: I would say: "… Feedback connections within an architecture of forward connections."
Line 487: after you say "we propose another hypothesis" could you summarise this final hypothesis in one sentence (e.g., my summary about the epigenetic specification of networks that functionally differentiate above).
I hope that these comments help should any revision be required
Barlow, H. B. (1974). " Inductive inference, coding, perception, and language." Perception 3: 123-134.
Felleman, D. and D. C. Van Essen (1991). "Distributed hierarchical processing in the primate cerebral cortex." Cerebral Cortex 1: 1-47.
Hilgetag, C. C., M. A. O'Neill and M. P. Young (2000). "Hierarchical organization of macaque and cat cortical sensory systems explored with a novel network processor." Philosophical Transactions of the Royal Society B: Biological Sciences 355(1393): 71-89.
Linsker, R. (1990). "Perceptual neural organization: some approaches based on network models and information theory." Annu Rev Neurosci. 13: 257-281.
Markov, N., M. Ercsey-Ravasz, D. Van Essen, K. Knoblauch, Z. Toroczkai and H. Kennedy (2013). "Cortical high-density counterstream architectures." Science 342(6158): 1238406.
Optican, L. and B. J. Richmond (1987). "Temporal encoding of two-dimensional patterns by single units in primate inferior cortex. II Information theoretic analysis." J Neurophysiol. 57: 132-146.
Rao, R. P. and D. H. Ballard (1999). "Predictive coding in the visual cortex: a functional interpretation of some extra-classical receptive-field effects." Nat Neurosci. 2(1): 79-87.
Srinivasan, M. V., S. B. Laughlin and A. Dubs (1982). "Predictive coding: a fresh view of inhibition in the retina." Proc R Soc Lond B Biol Sci. 216(1205): 427-459.
Author Response
Thank you very much for nice comments given by all of three reviewers, all of which were quite helpful for the revision of our manuscript.
The following is my reply to each comment.
Please see also an attached file that indicates the revised text explicitly.
For 3rd Reviewer
Comments and Suggestions for Authors
I enjoyed reading this interesting use of the Kolmogorov-Arnold representation theorem to account for the epigenetic specification of neural networks and their functional differentiation. I do not fully understand your hypothesis; however, it sounds intriguing. I suspect that you will need to unpack this hypothesis using numerical simulations before other people will understand the idea. However, as a thought provoking hypothesis piece, I thought the current submission was viable.
I have a few comments that might help unpack some of your assertions. Perhaps you could consider the following:
Line 133: I would qualify the assertion that most network structures in the brain are hetero. Perhaps something like:
“One could argue that the recurrent nature of connectivity in the human brain renders the architecture heterarchical. However, people distinguish between forward (ascending) and backward (ascending) connections that have distinct anatomical and physiological properties. This distinction has been used to organise brain areas into a traditional hierarchy (Felleman and Van Essen 1991, Hilgetag, O'Neill et al. 2000). However, there are a sufficient number of violations of a hierarchical designation to render brain architectures heterarchical. Perhaps the best example of this are the frontal eye fields that occupy a high and low level in an archetypal hierarchy, depending upon the connections in question (Markov, Ercsey-Ravasz et al. 2013)."
(Reply) I follow this very kind suggestion. I changed the original text in the following way (l. 131-150): 
The hierarchical system of this sense can be observed in deep neural networks following deep learning with the back-propagation algorithm. Another hypothesis proposes that hierarchically organized modular networks are evolved under the constraint of minimization of connection costs [44]. By contrast, actual brain networks typically consist of feedforward networks with feedback connections from a higher layer to multiple lower layers, whereby a simple hierarchy of function seems to be difficult to organize (see, e.g., [12] & [15]). In particular, the recurrent connections that occur in the brain could render the network architecture heterarchical, as stated by McCulloch. However, neuroscientists have distinguished between forward (ascending) and backward (descending) connections that have distinct anatomical and physiological properties, which may lead to distinct cognitive properties. This distinction has been used to integrate brain areas into a traditional concept of hierarchy [45–46], even in a dynamic phase [47]. However, a sufficient number of violations of a hierarchical designation exist: a small-world network, for example, which is actually observed in the cortex, makes it difficult to render the organization of brain areas hierarchical in a traditional sense [48].
Therefore, casting the idea of a “heterarchical” organization in the brain will be valuable for future studies of “hierarchical” organization. it can be considered that a hierarchical system in the McCulloch–Cumming sense is difficult to be self-organized in the brain. Most observed network structures in the brain are a heterarchical type.
Line 141: I think he you should define more clearly what you mean by stationary and far from equilibrium. I imagine that you meant something like:
"Non-stationarity in far from equilibrium systems refers here to itinerant dynamics over protracted timescales. Stationarity should be distinguished from steady state, in the sense that it is possible to have nonstationary itinerant dynamics in far from equilibrium systems with a steady-state solution. We mention this because what follows could be applied to (itinerant) systems that have a nonequilibrium steady-state solution."
(Reply) I inserted the following sentences (l. 159-162):
Here, stationary states are defined as unchanged probability distribution, and far-from-equilibrium systems are open systems, in which the states cannot reach the equilibrium states, but could reach steady (unchanged in time), periodic and chaotic states (see, e.g., [6–7]).
Line 200: could you replace "we adopted" with "we applied".
(Reply) Yes, I have done.
Line 209: I would say: "These numerical analyses suggest that a heterarchical structure is relevant for the effective…"
(Reply) Yes, I agree.
Line 323: similarly, replace "computation results" with "numerical analyses".
(Reply) OK
Line 324: it might be useful to reference related hypotheses. For example, the minimisation of errors underwrites predictive coding formulations of message passing in the brain (Srinivasan, Laughlin et al. 1982, Rao and Ballard 1999). You might also refer to formulations in terms of information transfer and maximum mutual information (Barlow 1974, Optican and Richmond 1987, Linsker 1990).
(Reply) Yes, I inserted some sentences with references in the legend of Fig.4 and a part just before Hypothesis 2 in the following way:
Fig. 4. Numerical evidence of the heterarchical structure of an ERC. (a) Change in the internal network structure consisting of an input and output layer from an initial random network to an evolved one. The network change proceeded with the change of wiring topology and connection weights, according to the optimization algorithm, such as the minimization of errors (present case, see also [2] for ERC and [39–40] for predictive coding formulations), minimization of energy cost, or maximization of information (see, e.g., [3], [37–38], [41–42]). The colors of the nodes indicate the degree of information quantity shared with the spatial or temporal output neurons: the higher the shared information with spatial (temporal) output neurons, the deeper the reddish (bluish) color of the node. The colors of the edges are as follows: red for feedforward connections; blue for feedback connections; green for connections within the input layer; and purple for connections within the output layer. The thickness of the lines indicates the magnitude of the connection weights. (b) Change in the accuracy of the realization of functional differentiation with the change in the scale factor of feedback connections. The red (blue) curve denotes the accuracy of the spatial (temporal) neuron.
Considering these numerical analyses and other works [37–42] computation results, we propose the following hypotheses.
Hypothesis 2. The functional differentiation for neuronal specificity, such as responding to specific external stimuli, evolved to minimize errors, which suggests the maximization of the transmitted information while reducing energy consumption.
Line 335: I would say: "… Feedback connections within an architecture of forward connections."
(Reply) Thank you !
Line 487: after you say "we propose another hypothesis" could you summarise this final hypothesis in one sentence (e.g., my summary about the epigenetic specification of networks that functionally differentiate above).
(Reply) I inserted one sentence (l. 555-557):
we proposed another hypothesis about the epigenetic specification of cellular indices that produces functional differentiation
